



# Optimized Merging of Search Coil and Fluxgate Data for MMS

David Fischer[1,5], Werner Magnes[1], Christian Hagen[1], Ivan Dors[2], Mark W. Chutter[2], Jerry Needell[2], Roy B. Torbert[2], Olivier Le Contel[3], Robert J. Strangeway[4], Gernot Kubin[5], Aris Valavanoglou[1], Ferdinand Plaschke[1], Rumi Nakamura[1], Laurent Mirioni[3], Christopher T. Russell[4], Hannes K. Leinweber[4], Kenneth R. Bromund[6], Guan Le[6], Lawrence Kepko[6], Brian J. Anderson[7], James A. Slavin[8], Wolfgang Baumjohann[1]

[1]Space Research Institute, Austrian Academy of Sciences, Graz, Austria
[2]Space Plasma/Magnetospheric Physics, University of New Hampshire, Durham, USA
[3]Laboratoire de Physique des Plasmas (UMR7648), CNRS/Ecole Polytechnique/UPMC/Univ. Paris Sud/Obs. de Paris, Paris, France
[4]Institute of Geophysics and Planetary Physics, University of California, Los Angeles, USA
[5]Signal Processing and Speech Communication Laboratory, Graz University of Technology, Graz, Austria[64]
[6]Goddard Space Flight Center, NASA, Greenbelt, USA
[7]Applied Physics Laboratory, John Hopkins University, Laurel, USA
[8]Climate and Space Sciences and Engineering, University of Michigan, Ann Arbor, USA

*Correspondence to*: David Fischer (david.fischer@oeaw.ac.at)

**Abstract.** The Magnetospheric Multiscale mission (MMS) targets the characterization of fine scale current structures in the Earth's tail and magnetopause. The high speed of these structures, when traversing one of the MMS spacecraft, creates magnetic field signatures that cross the sensitive frequency bands of both search coil and fluxgate magnetometers. Higher data quality for analysis of these events can be achieved by combining data from both instrument types and using the frequency bands with best sensitivity and signal to noise ratio from both sensors. This can be achieved by a model based frequency compensation approach which requires the precise knowledge of instrument gain and phase properties. We discuss relevant aspects of the instrument design, the ground calibration activities, describe the model development and explain the application on in-flight data. Finally, we show the precision of this method by comparison of inflight data. It confirms unity gain and a time difference of less than $100\,\mu s$ between the different magnetometer instruments.

## 1 Introduction

The MMS mission (Magnetospheric MultiScale, Burch et al., 2015) is comprised of 4 satellites that are used to measure plasma processes in the Earth's magnetosphere. The main mission target is the exploration of magnetic reconnection. To support measurements down to electron scales, the satellites are flying in a tight formation with distances down to 10 km. One of the main measurement quantities used to characterize plasma processes is the magnetic field, It is measured by 3 instruments which are part of the MMS FIELDS suite (Torbert et al., 2014): The analog fluxgate magnetometer (AFG), the digital fluxgate magnetometer (DFG) and the search coil magnetometer (SCM). As the measured plasma structures can have speeds from 10



to 1000 km s$^{-1}$, high measurement accuracy both in magnetic field magnitude and timing is required over a wide frequency range, e.g. to calculate speed and direction of a front crossing the tetrahedron satellite configuration.

The two fluxgate magnetometers are able to measure the magnetic field between DC and 64 Hz. The noise floor is around 5 pT $\sqrt{\text{Hz}^{-1}}$ at 1 Hz and it increases towards lower frequencies (Russell et al., 2014). The frequency responses of both fluxgate

magnetometers are flat for lower frequencies and corresponds to a low pass with low filter order and a corner frequency around 30 Hz. Therefore, fluxgate data are typically used for scientific analysis without any compensation of the frequency dependent gain and phase characteristics.

The search coil magnetometer is suitable for measuring much higher frequencies, but its sensitivity at lower frequencies is limited by the underlying induction principle. Its frequency response represents a transfer function of higher order and requires

compensation based on ground calibration measurements (Le Contel et al., 2014).

A noise floor comparison, which partly reflects the frequency response, is shown in Figure 1. The plot compares the digital output of the DFG with the analog output from the SCM. The frequency range of equal noise floor is between 3 and 6 Hz, although this may vary across different instrument implementations. With the wide frequency range of plasma events mentioned above, it is desirable to have a common merged product that combines the best data of both instruments, i.e. all

available frequency bands with lowest possible noise floor. This is particularly useful for observations of electron diffusion regions and thin current sheets which feature signatures in the frequency range from 0.5 to 20 Hz (Torbert et al., 2014).

A similar approach was already used on magnetic field data from the Cluster and Themis missions, but in both cases it was based on limited on-ground frequency response calibration and usage of in-flight data for calibration. The SCM frequency characteristics were in principle known (Cornilleau-Wehrlin et al., 1997; Roux et al., 2008), but the knowledge of absolute

time was limited. The knowledge of the frequency response of the fluxgates (Balogh et al., 1997; Auster et al., 2008) was limited to discrete frequency points in amplitude and low accuracy on end-to-end phase and timing information. Merging for data analysis was therefore only possible using this limited data and in-flight comparison (Alexandrova et al., 2004). Furthermore, comparative calibration of the SCMs was done by matching to fluxgate data at very low frequencies, which are not influenced by the fluxgate low pass characteristic. (Robert et al., 2014).

Only the precise knowledge of the magnitude and phase response of both instrument types allows for accurate merging of the respective data, keeping intact phase and gain relations between different frequency components of the observed magnetic signatures. Thereby the shape of the signatures is preserved in the merged data product, as higher frequency components of steeper slopes stay phase aligned. Hence, the merged data are well suited for the analysis of internal fine structures of magnetic signatures as well as for high precision determinations of dipolarization front speed and direction by multi-spacecraft timing

analysis. In addition, the comparative calibration of gain and alignment between SCM and AFG/DFG is massively improved in flight. Without the precise knowledge of their frequency responses either data with low signal to noise ratio (SNR) from the SCM (i.e. at the satellite spin frequency) or data with little phase knowledge from the fluxgate magnetometers have to be used for comparative calibration. This results in alignment errors and inaccurate gain factors. With this knowledge, data from the complete common frequency band can be used and areas with good SNR can be selected.



The MMS FIELDS team invested significant efforts in instrument design and test to ensure that end-to-end timing and frequency response information is already available from on-ground calibration. Design efforts include a common master clock and synchronization, which enables sampling of SCM, AFG and DFG with constant phase relation, as well as high precision time stamping. The target of these efforts was to reach an absolute time tagging accuracy of 100 µs across the full bandwidth,

despite the heterogeneous architecture of the instruments. This required common clocks and synchronization, as instruments are sampled by different means. AFG and DFG are sampled by their own electronics, whereas the SCM is sampled by the digital signal processing (DSP) unit (Ergun et al., 2016). In both cases, depending on downlink data rate, also digital filters for sample rate conversion are applied, which also needs to be taken into account for time tagging.

This work deals with the method to create a merged 1024 Hz data product from 128 Hz fluxgate and 8192 Hz search coil data.

In a first step this required the verification of absolute timing and the identification of the instrument's frequency responses in an end-to-end test on ground. In a next step data from these tests have been used to create instrument models, which can be employed for inverting the instrument's frequency responses and to make the data fit for precise merging. In the final part, in-flight data were merged using these models and the results were evaluated to verify the timing precision of the complete process.

## 2 Approach

Characterization of an unknown system is done by using known input signals, measuring the system response and creating a model using both signals. For magnetometers the input signal has to be a magnetic field stimulus and the system response is the digital measurement result. Since both instrument types can be considered as linear and time-invariant for the case of a merged data product, it is possible to describe them with a model based on finite (FIR) or infinite impulse response filters

(IIR). A suitable magnetic field stimulus can e.g. be created by a solenoid coil system. Amplitude and phase of the magnetic field are linked to the applied current.

### 2.1 On-Ground Calibration and Measurement Setup

Figure 2 shows a block diagram of the setup used for the frequency response calibration. All magnetic field sensors of the FIELDS instrument suite were placed in a mu-metal can and the stimulus signal was applied using a current generator and a

solenoid coil. AFG and DFG were digitized by the respective front-end electronics, while SCM signals first passed through a preamplifier and were then sampled by the DSP unit. The data streams were then processed by the FIELDS central data processing unit (CDPU).

The current generator and the CDPU were synchronized by a common 1 Hz time reference clock. All data recorded within FIELDS were time stamped relative to this reference. Furthermore all instruments within FIELDS were synchronized using

common master and synchronization clocks. A more detailed description of this mechanism is available in the FIELDS instrument paper (Torbert et al., 2014).



The current generator that was constructed for this test is able to drive an arbitrary waveform current. It is generated by an internal digital signal generator. This signal generator is operated at a sampling frequency of 4096Hz which is synchronized to the reference. This sampling clock is derived from a 67 MHz master clock internal to the current generator, but instead of using a fixed divider, additional clock cycles are added (or subtracted) as required to keep it synchronous to the 1Hz time

reference. These added clock cycles must be considered as artificial jitter which in principle creates additional noise in the stimulus signal. The maximum amplitude of this noise can be calculated by using the maximum possible signal change within the maximum jitter time. For a jitter of 15ns and a maximum signal frequency of 1024 Hz this results in a signal to noise ratio of approximately 80 dB, which was sufficiently low for this purpose. As every real current source will produce small differences between desired and achieved current, an additional current measurement was mandatory. A comparison of the

generated current and the 1 Hz time reference signals shows a time deviation in the sub-microsecond range.

The current stimulus was driven through a solenoid coil in a mu-metal shielding, which attenuates external fields during the measurement. The influence of this coil/can system is minimal in the frequency range of interest from DC to 512 Hz, therefore the generated field could still be considered as proportional to the measured current.

The stimulus waveforms used for testing needs to cover the whole frequency range of interest. Commonly used test signals for

identifying unknown frequency responses are discrete frequency sines (using interpolation in between), sine sweeps and noise signals. Discrete and sweep sine signals have the advantage that the signal frequency for any given time is known and therefore all signals with different frequencies can be removed by bandpass filtering. On the other hand, longer measurement times are needed to excite all frequencies. Noise measurements excite all frequencies within a short period of time, but this method is sensitive to all additional noise sources within the analyzed frequency band. However, any uncorrelated noise can be reduced

by longer measurements and averaging.

For the FIELDS frequency response calibration three different tests were conducted with each of the instrument suites of the 4 MMS satellites. In a first test, the 1 Hz time reference was connected to one of the additional voltage channels of the DSP. In a second test, the instrument responses were measured only at a few discrete frequencies with sine stimuli. In a third test, pink noise with a bandwidth from DC to 1024 Hz was applied to all sensors and the instrument response was recorded. The

choice of pink over white noise was driven by the sensitivity curve of the SCM, as the higher frequency components of a white signal would drive the SCM to saturation even when the measurement output at lower frequencies is small.

In addition, a voltage fully represented to the current stimulus was connected to a DSP voltage channel. This way, the current was also recorded synchronously to the FIELDS clock and so it could be used in later stages without the need for further resampling or time shifting. Additionally, this resulted in faster data transfer and easier data formatting.

**2.2 Results of on-ground Calibration**

Using data from the first test with, which was the digitizing of the time reference signals, it was possible to measure the delay introduced in sampling and time stamping of the DSP voltage channels by comparing the sampled waveform of the reference clock and its time stamp. As the 1Hz time reference is giving the reference for the beginning of a full second, the clock edge



within the sampled waveform should ideally be time stamped with this second. The measured deviation from the expected delay was less than 7 µs.

The results of the second test based on the sine signals delivered a highly accurate measurement of gain and phase at discrete frequencies and an initial approximation of the frequency response, both for the magnetic field instruments as well as for the current measurement via the DSP voltage channel.

These first two tests showed, that the DSP voltage channels had sufficiently flat frequency response and that the knowledge on timing accuracy was better than one microsecond. The later noise tests were therefore conducted using the current measurements of the DSP channels, as this resulted in reduced effort in calculations. The additional delay of these channels was accounted for in all further tests. For verification purposes, both the sine and time reference measurements were included

in all further test series, to ensure that no changes occurred in the setup.

The resulting data of the third test with applied noise signals were used to create an initial estimate of the full instrument frequency responses. This estimate can be found by dividing the FFTs of stimulus and instrument measurements, which is the frequency domain equivalent of deconvolution.

Unfortunately, this method is not necessarily delivering an estimate that is fully representative for the system. The FFT method

implies that a single transform window only contains the convolution of the transfer function and a stimulus. In reality the beginning of each FFT window contains a part of the response to previous stimulus data and a part of the response to the current stimulus is truncated. In addition also noise and distortion within the window are considered as part of the frequency response. The problem of previous as well as truncated response data can be minimized by choosing a sufficiently large FFT length and by using windowing, overlapping and averaging. The problem of measurement noise is more complicated.

Although normal noise can be reduced by averaging, this is not the case for systematic distortion like power line tones (50/60Hz and harmonics). In this case the resulting estimate would have a changed frequency response at the respective frequencies. The FFT based estimate can therefore not be used as a model.

Figure 3 and 4 show examples of both sine and noise based frequency response estimates results from the Y axis of flight models 4 (used on MMS 3). Figure 3 shows that AFG and DFG have the expected low pass characteristics. The DFG has a

higher corner frequency and constant phase delay. The noise visible in the phase delay plot results from the translation of phase noise to phase delay, which increases the noise at lower frequencies. The found delays for DFG match the expected digital delay with a maximum deviation of 30 µs. The delay and gain curve for other axes of the AFG have a slightly different characteristic due to the differences in analog elements.

Figure 4 shows the lower end of the expected SCM bandpass characteristic, which also matches the reference measurements

taken by the SCM team in Chambon-la-Forêt (Le Contel et al., 2014). The frequency response for higher frequencies was not measured, as this was not required for the merged 1024 Hz data product.

The absolute timing of the FFT estimates was calculated by adding the delays of the DSP voltage channels known from the first two tests. A comparison between FFT estimate with added delay and the sine measurements, which do not include the



current measurement via the DSP channel, shows good agreement and is therefore verifying the correct handling of delays in the DSP voltage channels.

## 2.3 Model Development

The SCM model was chosen according to a theoretical model from the SCM team (Le Contel et al., 2014). As parts of the

frequency transfer function of this model are far above the frequency band of interest, the model was reduced to a second order IIR filter based model and a fractional sample Lagrange delay. The IIR model was then optimized to fit the measurement results in frequency domain. Inversion of this model is not directly possible, as it is not minimum phase (Oppenheim, 1999) – which means that the inverted model is unstable and has poles outside of the unit circle. This is also visible by the differentiating feature of the SCM. If this feature would be inverted, the resulting integrator would produce infinite values

even with small DC inputs. As the lower parts of the spectrum are not used in the later merged product, it is possible to design a similar IIR transfer function with nonzero DC gain and minimum phase property by just adding a small additional coefficient, thus changing the original numerator polynomial to have all poles within the unit circle. The resulting inverse model is a low shelving filter with a DC gain of 220 dB, which is close to the unmodified model for all frequencies above 0.1 Hz.

The main contributor to the frequency response of the DFG is the digital averaging filter used for the DFG internal

downsampling. (Magnes et al., 2003) Also this filter cannot be considered as minimum phase and does not allow direct inversion. However, since only the spectral part below 64 Hz is used in the final 1024 Hz product, the properties of an inversion filter are irrelevant above these frequencies. This fact leaves some degrees of freedom for an optimization solution. The basic model in this case is a 128[th] order FIR model that was derived using a Wiener-Hopf solution (Haykin, 2002). This method delivers a model that converts the input stimulus to a model output signal which is tracking the real instrument measurements

with minimum mean square error.

For the DFG this principle was inverted by exchanging input and output, thus creating an inverse model that can directly be applied on instrument data to reconstruct the "original" magnetic field data. Additionally, the instrument measurement data are compensated for the delay before modelling, as otherwise the model would have to resemble both delay and frequency response of the instrument. Keeping these delays within the model would in principle only cause a shift of the model

coefficients by adding zero coefficients at the beginning. This addition would increase the filter order that is used for optimization and instead of keeping these coefficients at zero, an optimal solution would instead try to model just the instrument noise to get to a minimum error. Introducing a time shift is therefore reducing model order and is avoiding coefficients that just model the noise.

The same approach, although with different delay compensation, was used on the AFG data. In this case the characteristic part

of the filter is the analog low pass of the feedback regulation loop, which also introduces non-constant group delay. The respective modeling process of each instrument was applied individually to each of the instrument axes,

The found instrument models were transformed to the frequency domain, inverted and compared to the FFT and sine based frequency responses. Figures 5 to 7 show the differences between FFT based results, model response and reference calibration





(Le Contel et al., 2014). The SCM model comparison in Figure 5 shows the presence of powerline noise as peaks in amplitude and phase, both in the FFT (60Hz) and the reference (50Hz) measurements. These peaks are absent in the models and are therefore visible as differences in the comparison. The constant delay between reference and FFT as well as model measurements is due to the fact that the SCM reference calibration was only done using the analog sensor output and does not

include all digital delays introduced by sampling.

Data from DFG and AFG presented in Figures 6 and 7 show a very good match between model and FFT based frequency response measurement results and, for DFG, also with the theoretical values of the digital averaging filter. The remaining delay variation of around 20 µs is well below the 100 µs goal. It is caused by the limited length FIR implementation, which could be corrected by a filter of higher order – but this has disadvantages, as shown in the next paragraphs.

The last step of model creation was to normalize the models to the results of regular on-ground gain calibration of the instrument teams. For the fluxgates this was done by setting the DC gain to one, so DC calibration would be unchanged. For the SCM it was done by matching the gains of model and on-ground reference calibration around 1kHz.

The filter based instrument models require an initial filter settling time that is dependent on the filter impulse response length. Data from this period needs to be removed from the final data product and is therefore producing a data gap. This is of special

concern for MMS high sampling rate data, as only short data bursts are transmitted and a loss of many data points cannot be tolerated. Some of this could be mitigated by prefilling with data from lower sampling frequencies, but this process is complex, as multiple frequency response compensation is involved. The best solution is therefore the use of limited length filter functions.

This is already the case for fluxgate model filters, but the IIR characteristic of the SCM requires in principle one more iteration.

For later merging, the compensated SCM data is filtered with a high-pass filter. These two filter operations for compensation and merging can be combined to a single filter by convolution. The impulse response of this convolved filter is theoretically infinite, but practically decaying very fast, e.g. using a 1025 point merging filter (see below) the impulse response decreases to $10^{-13}$ after roughly 1000 points. Numbers of this size are far below the instrument noise and can therefore be neglected. It is therefore possible to replace the IIR filter by its truncated impulse response without relevant changes in the frequency response.

For this paper, the IIR filter was used as SCM model. A replacement by an FIR model is planned in the future.

Data from both instruments is merged with a crossover filter that is weighting different spectral parts of the instruments according to their properties. An optimum crossover filter set would track the exact minimum noise level of the instruments. This is difficult to realize, as the ability to track the best noise floor is directly connected to the order of the filter and will therefore again cause data loss due to filter settling. We therefore chose to implement a windowed FIR low and high pass filter

based on the sinc function, which also provides the advantage that design of the respective complementary filter is a simple subtraction from the Dirac delta. It means the sum of the two filters is unity gain. Furthermore, comparison to unfiltered signals is simple due to its constant group delay property.

Also here the window length has to be matched to an acceptable data loss due to filter settling. With larger window size stopband attenuation increases, passband ripple decreases and the crossover characteristic has a steeper slope. Figure 8 shows



an example of two filter sets with different window lengths and the resulting differences in attenuation and crossover slope. For initial merging and comparison the 16385 point filter was used, but in later mission phases with shorter data bursts this will be changed to 2049 points or less, depending on available burst data length.

## 3 Application

The final process of merging, which is suitable for automated application, is shown in the data flow diagram in Figure 9. Apart from the already discussed crossover filters and model based frequency response compensation, a few more blocks are present in this diagram. The uncalibrated data files from all instruments (called L1A data according to MMS definition) is passed through a block that handles fragmentation, as data files can have gaps and contiguous data can be distributed over several files.

The resampling block includes antialiasing filters and converts the different data products to the final product rate of 1024 Hz. The first remaining sample in the decimated data product of SCM is selected by looking for the closest neighbor in the fluxgate data, thus minimizing the time distance between those samples. This reduces the amount of time shift needed to synchronize the data to a common sample time basis.

The blocks for timestamp updates and fractional delays are two separate parts of a common mechanism. Fractional delays (less

than a sampling period) are required to align data to a common time basis and to compensate absolute time delays. A pure fractional time shift can be achieved by interpolation, but would require infinite data. All realizable methods use limited time interpolation which affects gain and phase at higher frequencies.

In the time stamp update block only the time stamp is corrected by the known delays, while in the fractional delay block the fluxgate data are interpolated to the time line of the search coil data. This way all time shifts are applied with a single fractional

delay filter and the influence on gain and phase is minimized. This shift was implemented as Lagrange filter and its overall spectral influence is minimal, as the higher frequency parts of the fluxgate spectrum are not used in the final merged data product.

Fluxgate data is undergoing regular in flight calibration (Russell et al., 2014) for orthogonality, alignment, offset and gain. The required parameters for this calibration are provided by the magnetometer team and are calculated using both on-ground

information as well as in-flight parameter adaptation. Furthermore, both data sets need to be transformed to a common coordinate system for merging. The chosen coordinate system for MMS is the orthogonal mounted boom system (OMB). The sequence of frequency response compensation, regular calibration and coordinate transforms is of importance. If coordinate transformation and orthogonality calibration were done first, this would result in a mix of data from different axes and therefore different frequency responses. For small rotations this error could be neglected, as in this case minor differences between these

responses would be scaled by the sine of small angles. Still, for larger rotations (e.g. boom to spacecraft body coordinate system), this error is larger and frequency response compensation should be done beforehand. The best way is of course to



apply frequency compensation as first step. Coordinate transforms are relying on already corrected gain and orthogonality, so those need to be done after the frequency compensation.

After merging, the data is finally transformed to the target coordinate system, i.e. Geocentric Solar Ecliptic (GSE) or Magnetospheric (GSM).

# 4 Evaluation

The merged data of about 2 months was used for evaluation of the data product, as tracking of small scale differences in time, alignment and gain required statistics on high rate burst data, which was not available at all times due to operational constraints. A first time domain comparison (Figure 10) in the band from 4-64 Hz shows good visual agreement between compensated data from SCM and DFG.

A more accurate analysis of the quality of the compensation models can be achieved by comparing relative gain and phase between compensated SCM and DFG data. The result of this comparison is shown in figure 11. The calculations for these figure was done by dividing the FFT spectra of the individual axes, which should ideally result in unity gain and zero phase for a system with perfect frequency response compensation. Data were analysed using FFT windows with a length of 2048 points and averaging over 10 minutes. Only data sets with relevant amplitudes in the frequency range 10-64 Hz were taken

into account, i.e. sets that have more than 10000 points above a 100 pT magnetic field threshold with an average of at least 150 pT.

The gain plot in figure 11 shows a gain factor that is close to unity up to at least 30 Hz. A clear interpretation above this frequency interpretation is difficult, as the noise increases massively. This is due to the low amplitudes of natural signals in this frequency range. These signals barely exceed the noise floor of the fluxgate, thus resulting in a poor signal to noise ratio.

The phase plot does not show a significant trend, but suffers from the same noise problem as the gain plot. Still the quality of timing determination is in this case better, as a timing error would result in a linear phase trend. For comparison purposes the time accuracy goal of 100 µs was converted to linear phase trends and added to the plots. Comparing this limit to the measurement result shows no linear trends in this order of magnitude and verifies that time stamping accuracy is definitely better than 100 µs in the investigated frequency band.

In addition to timing, also alignment and gain was compared by minimizing the differences between DFG and SCM measurements using linear combination of the different axes with a 3x3 alignment matrix. The result showed that the angle mismatch between DFG and SCM was in the order of 0.5 to 1°. This fits well to the expected differences, as SCM initial calibration assumes perfect orthogonality and alignment, while fluxgate data has full alignment and orthogonality calibration in place. The result of these comparison will be added to the SCM calibration flow in the near future.



## 5 Conclusion

The common effort of the MMS FIELDS team allowed to create frequency response models for the FIELDS magnetic field instruments that are based on full end-to-end on-ground calibration. Using these models good agreement between data from search coil and fluxgate magnetometers was achieved and data could be merged to a common product with a timing precision better than 100μs. Furthermore, the developed methods are suitable for automated processing.

With the frequency response compensated data also alignment and gain corrections for the search coil were generated by comparison with the in-flight calibrated fluxgate data. The resulting merged magnetometer provides a new basis for analysis of scientific events which contain frequencies ranges that are spread across two instruments.

## 6 Acknowledgments

The dedication and expertise of the Magnetopheric MultiScale (MMS) development and operations teams are greatly appreciated. Work at JHU/APL, UCLA, UNH, and SwRI was supported by NASA contract number NNG04EB99C. The French involvement (SCM) on MMS is supported by CNES and CNRS.

We acknowledge the use of burst L1A data from digital and analog fluxgate magnetometers and search coil magnetometers. This data is stored at the MMS Science Data Center https://lasp.colorado.edu/mms/sdc/ and are publicly available. The Austrian part of the development, operation, and calibration of the DFG was financially supported by rolling grant of the Austrian Academy of Sciences and the Austrian Space Applications Programme with the contract number FFG/ASAP-844377.

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



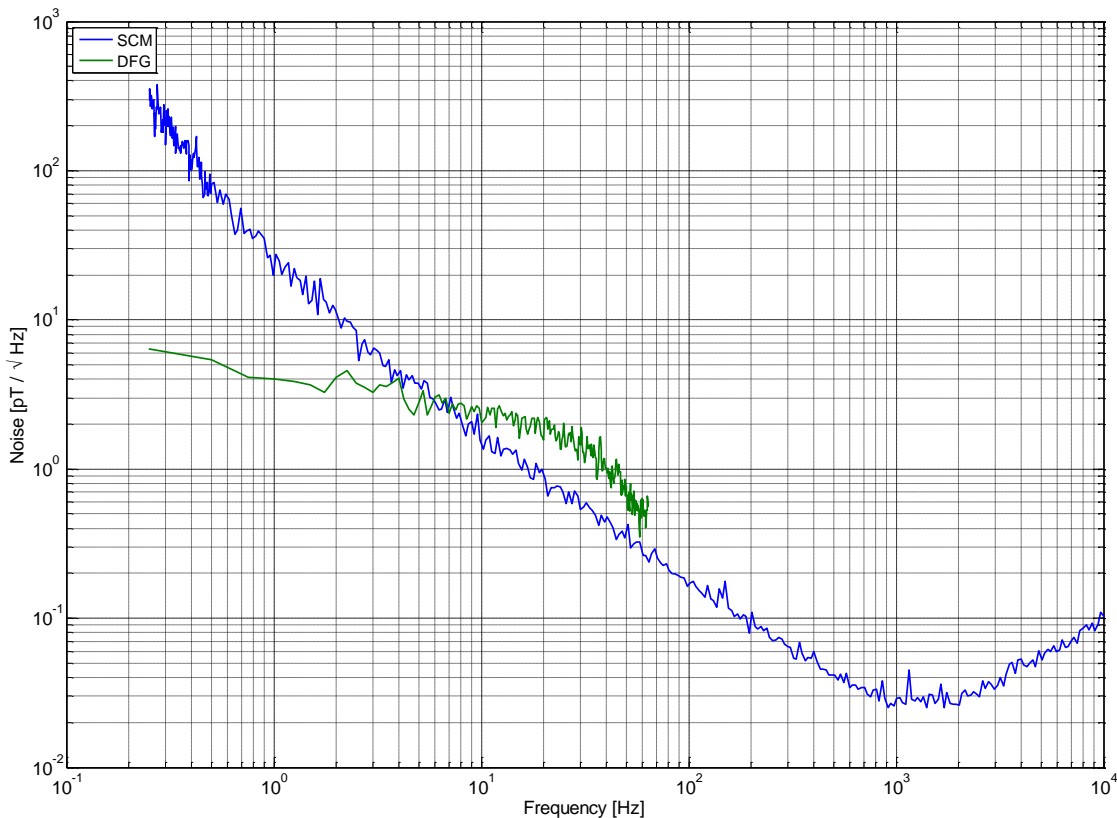

**Figure 1: Comparison of search coil (SCM) and fluxgate (DFG) noise floor**





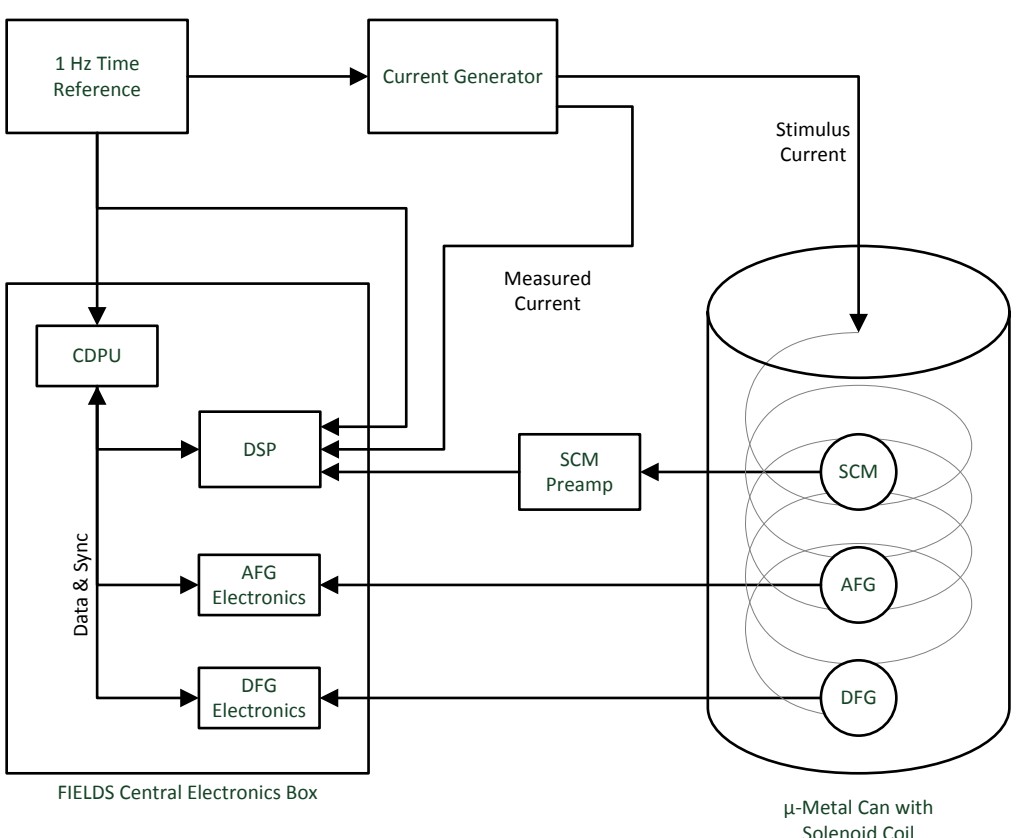

**Figure 2: Test setup for instrument frequency response measurements**





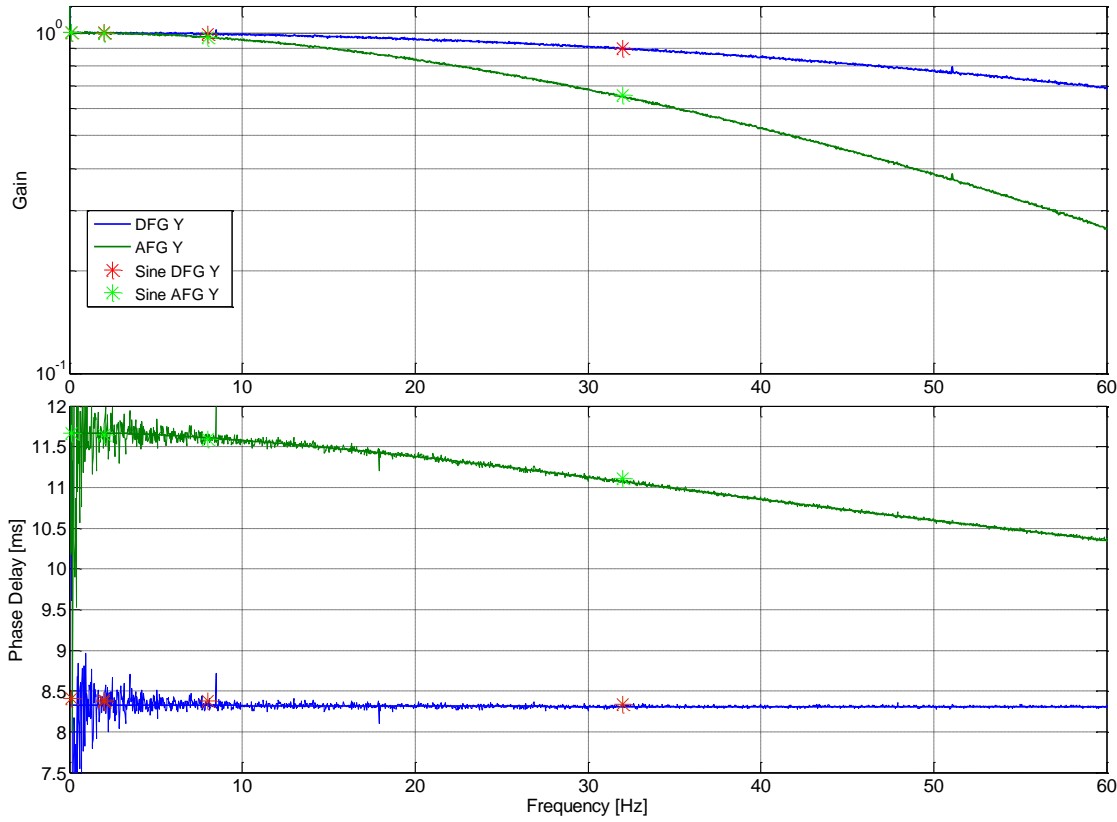

**Figure 3: Frequency response of AFG and DFG measured with FFT estimation method and sine signals**





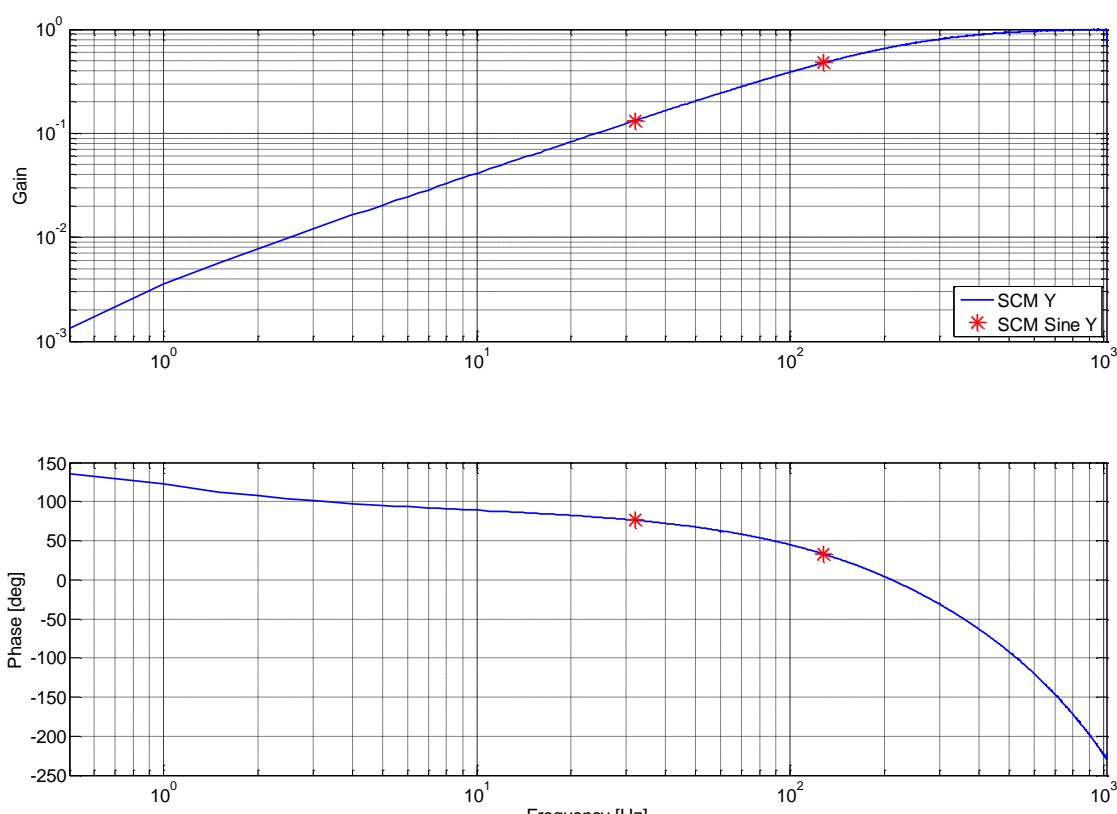

**Figure 4: Frequency response of SCM measured with FFT estimation method and sine signals**





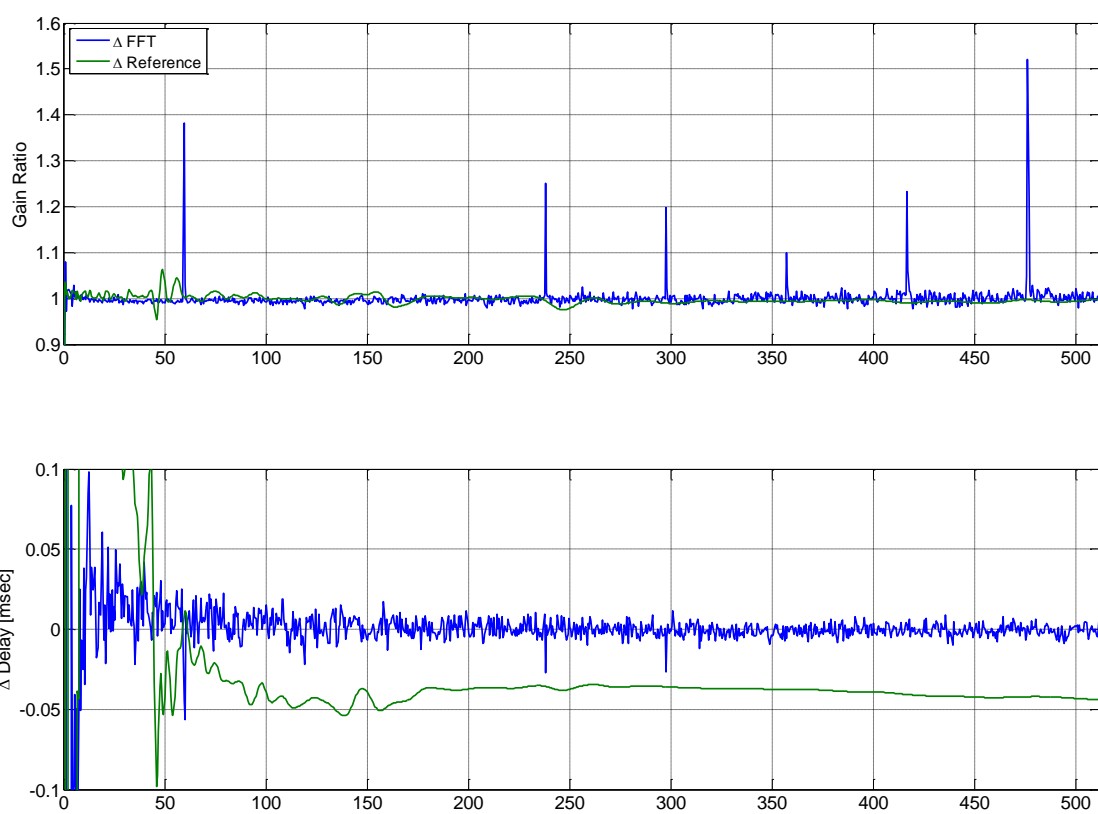

**Figure 5: Gain ratio and delay difference between SCM frequency response model and FFT estimate as well as analog reference calibration**





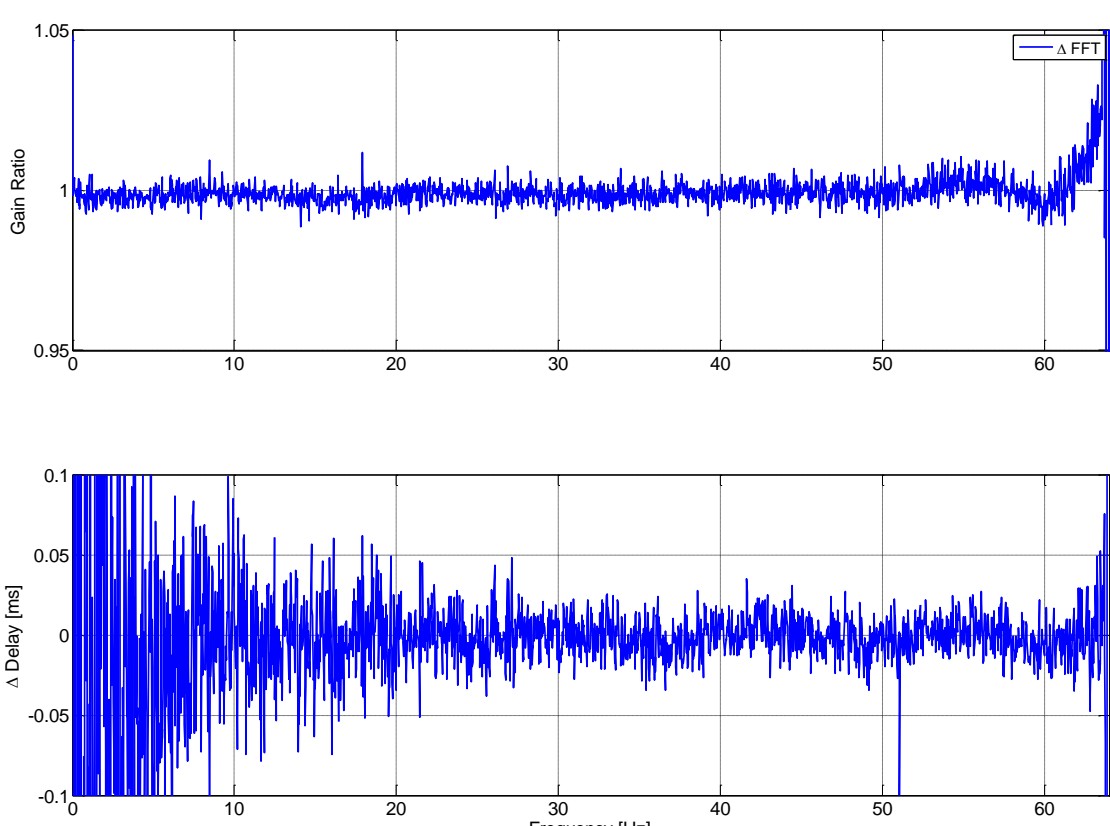

**Figure 6: Gain ratio and delay difference between AFG frequency response model and FFT**

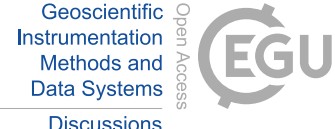



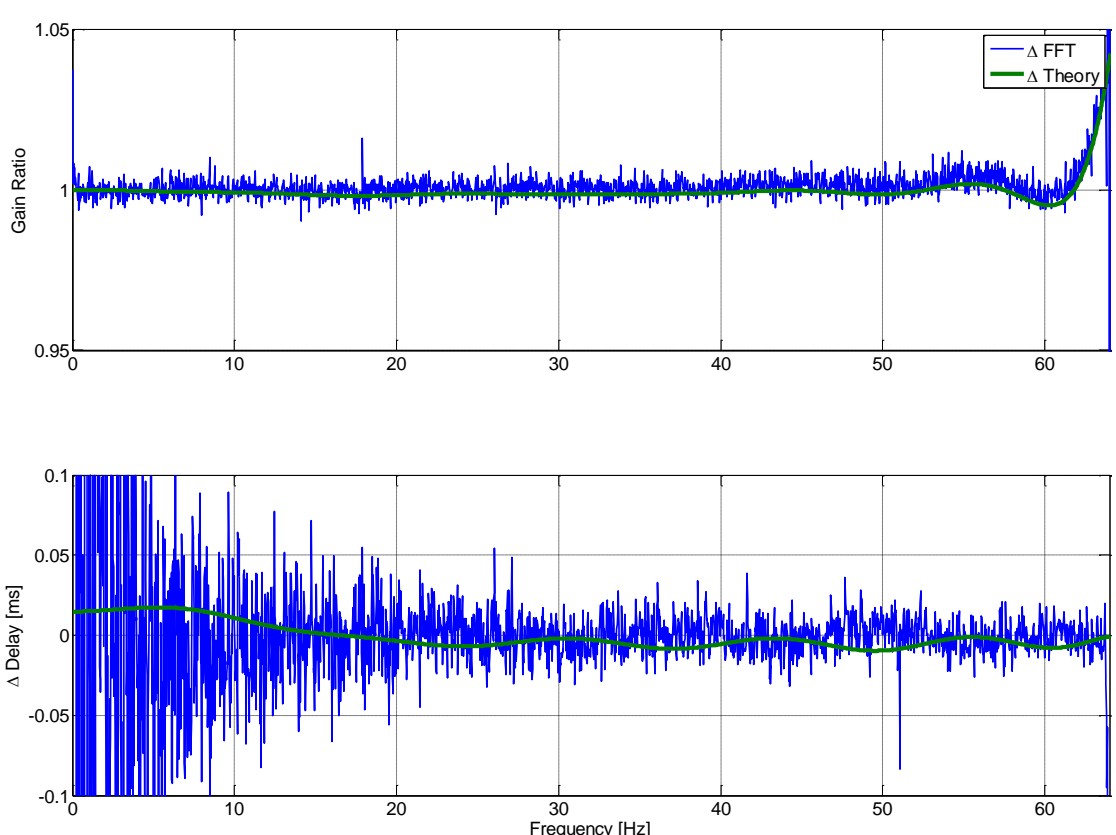

**Figure 7: Gain ratio and delay difference between DFG frequency response model and FFT estimate as well as theoretical digital filter response**





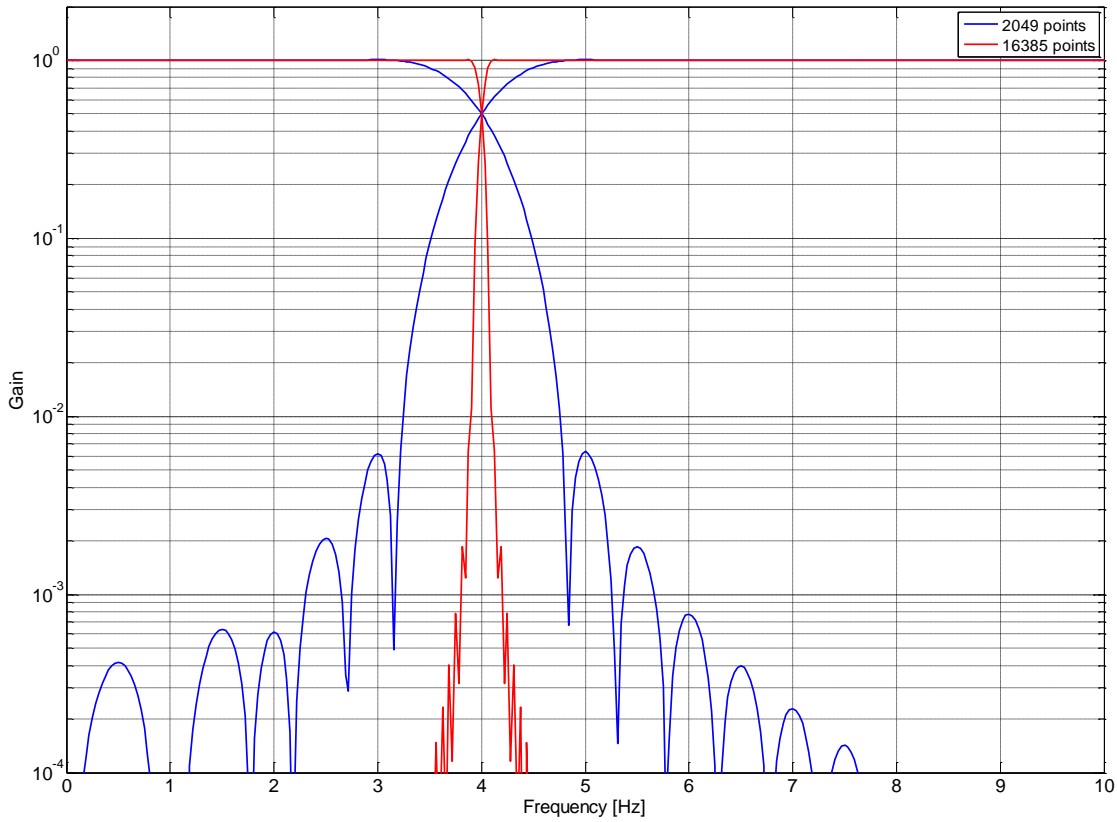

**Figure 8: Frequency response of two merging filters with different filter lengths**



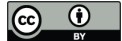

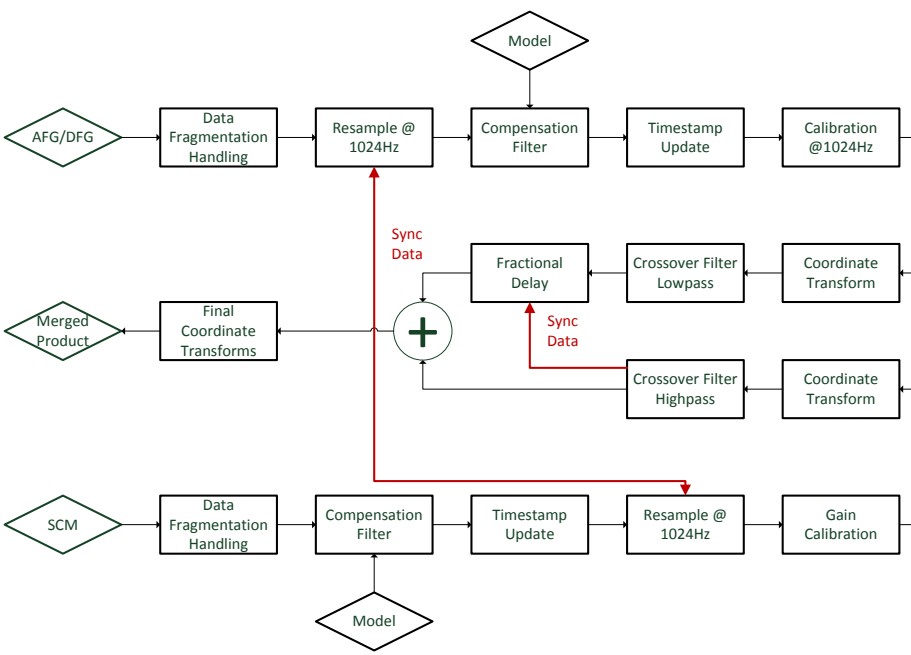

**Figure 9: Data flow block diagram for merging**

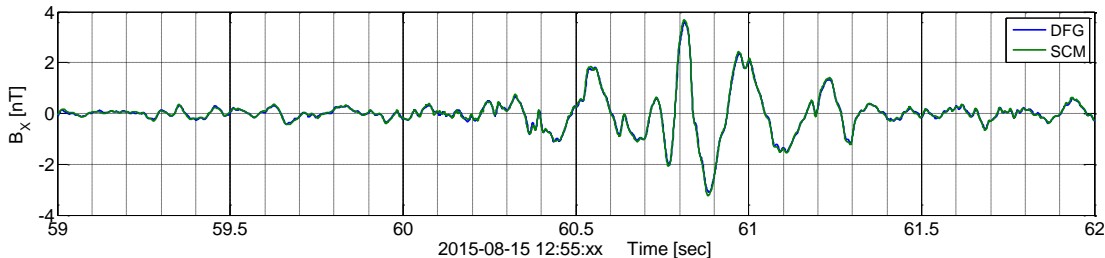

**Figure 10: Comparison of MMS 3 frequency compensated data of DFG and SCM in time domain with band limitation from 4 to 64 Hz**





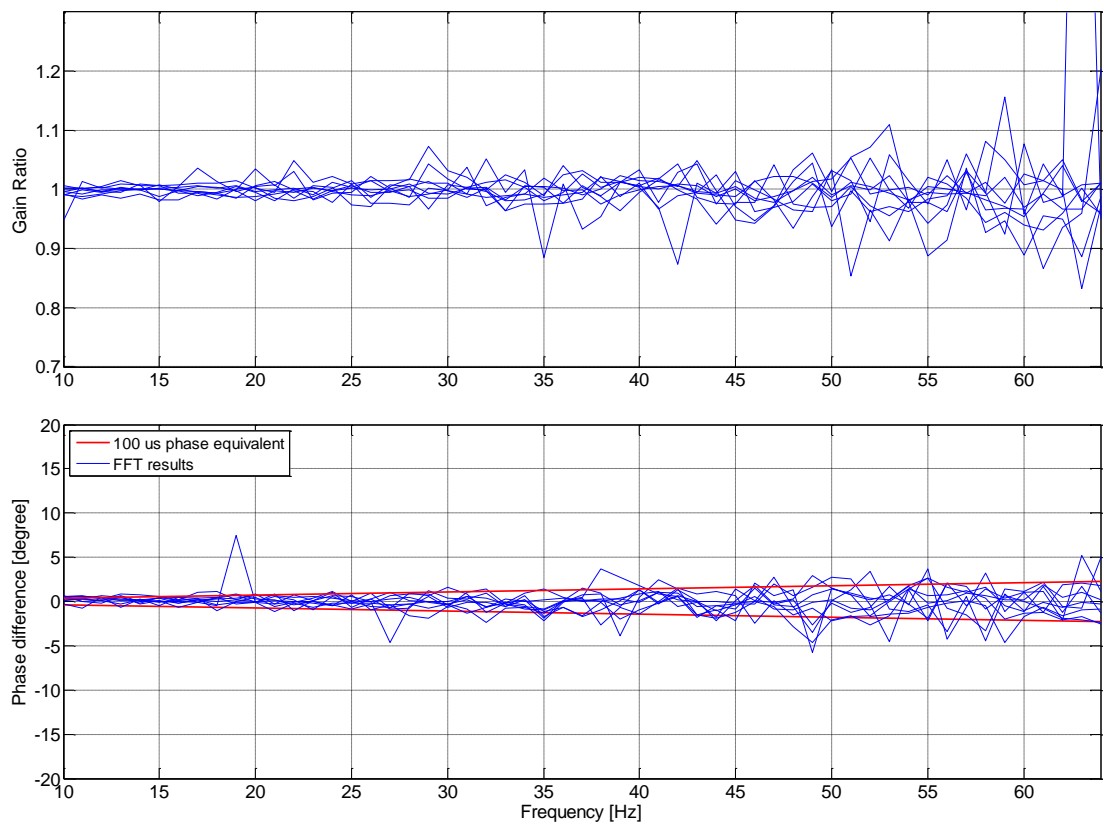

**Figure 11: Comparison of gain and phase for MMS 1 SCM and DFG data**