# Peer review of "Optimized Merging of Search Coil and Fluxgate Data for MMS"

_Geoscientific Instrumentation, Methods and Data Systems, 2016_

## Referee Comment (RC1) · Anonymous Referee #1 · 1 Jul 2016

**Referee report on the manuscript "Optimized Merging of Search Coil and Fluxgate Data for MMS" by D. Fischer et al.**

**General comments**

For measurements of some quantity the instruments of different types are often used. Particularly in space-born projects as well as during ground-based geophysical surveys the magnetic field fluctuations are simultaneously measured by fluxgate and search coil magnetometers. The optimal combining data of both instruments is important task for obtaining high quality results with the best available signal-noise ratio (SNR). From this point of view the approach, methods, models and techniques described in the paper are important not only for a case study devoted to the Magnetosperic Multiscale mission (MMS), but for a wide range of similar applications. Surely the paper tries to answer questions within the scope of GI journal.

In general the approach used by authors to merge data from the instruments with different frequency responses is straightforward and clear. It contains following steps. Basing on the noise characteristics specify the merging frequency, where noise levels of both magnetometers are equal. Then estimate the end-to-end frequency responses of individual instruments and develop its appropriate models. Apply these models for the frequency compensation of the magnetometers data, convert data to the same sampling frequency and synchronize samplings. Make alignment of the axes and gain corrections and, finally, apply appropriate crossover filters and merge the data taken in the appropriate for each magnetometer frequency bands.

The presented in the paper example of this approach application confirms that merging in-flight data were synchronized within required limits (<100 μs), however the estimation of the gain mismatch was quite large due to the low signal-noise ratio in the analyzed frequency band 10-50 Hz. Probably, a higher signal-noise ratio and better validation of the proposed merging procedure could be obtained, if the records of the noise signals used for the frequency response calibration would be compared instead or in addition to the in-flight data processing. The estimation of the noise level of the merged fluxgate and search coil data is highly recommended, as it would also confirm achieving of the main goal of this work – improving SNR of the combined data. In my opinion, the step and impulse responses of the merged data have to be checked too, in order to estimate the possible differences between the combined data and the original magnetic field signatures during the fast changes of the latter.

The important task is a selection of the proper parameters of the crossover filter for merging data. The authors states that a high order crossover filter is necessary in order "...to track the best noise floor..." of the instruments, however, this statement was not backgrounded in the paper and no appropriate references were given. First of all, it is unclear how the crossover filter could track the instruments' noise floor. Secondly, in my opinion, even the second order crossover filter is sufficient for combining data.

The overall presentation is quite well structured and clear, however, in some cases using mathematical expressions and formulae instead of text descriptions, as well as giving more details about the developed models and data processing techniques would be more useful for reproduction of the proposed approach by other researchers.

**Specific comments**

p. 6, section 2.3 Model Development.
For the sake of traceability of results the developed IIR model of SCM would be presented by expression in analytic form, and the typical shape of the impulse response of the FIR models for fluxgate magnetometers would be given.

p. 4, lines 27-29 and p. 5, lines 3-7. Was the current in the coil measured by an independent instrument? If not, it is not clear how accurately the frequency response of the DSP voltage channels was estimated. Is it assumed that the current amplitude at all applied frequencies was constant?

p. 5 lines 14-22. What were parameters of FFT-based frequency response calibration - duration of the stimulus signals, FFT length, overlapping, type of the correcting window, if any? What are principal limitations, which did not allow improving phase delay estimations at the frequencies < 20 Hz?

p. 6, lines 12-13. It is unclear why so high DC gain (220 dB) of the shelving filter is necessary for frequency compensation of the SCM transfer function in the band from 0.1 Hz to 500 Hz, while the ratio of the gains of the original transfer function at these frequencies is approximately equal to -86 dB (Figure 4 in the manuscript and Figures 9 and 14 in Le Contel et al., 2014).

**Technical corrections**

p. 2, line 4, in my opinion, it is better to use "...5 pT/$\sqrt{Hz}$..." or "...5 pT/Hz$^{1/2}$..." or "...5 pT Hz$^{-1/2}$..." instead of "...5 pT $\sqrt{Hz}^{-1}$ ..."

p. 4, line 31, probably word "with" could be omitted.

p. 5, lines 7, 8. The sentence "The later noise tests were therefore conducted using the current measurements of the DSP channels, as this resulted in reduced effort in calculations. " What does the phrase "the current measurements of the DSP channels" mean? Does it mean that during the noise tests the voltage proportional to the stimulus current in the coil was measured using one of the DSP voltage channels? In this case, probably, it has to be formulated as follows "the current measurements **via** the DSP channels".

p. 6, line 26, probably, the second word "instead" could be omitted.

p. 9, lines 17, 18, probably, the second word "interpretation" could be omitted.

---

## Referee Comment (RC2) · Anonymous Referee #2 · 8 Aug 2016

The combining data which is simultaneously measured by fluxgate and search coil magnetometers is important task for obtaining high quality results. So the tasks which authors solved are important for different applications. After correction the method of merging of search coil and fluxgate data became clear. Authors confirm that merging in-flight data can be synchronized within 100 mks. The unclear question is the noise level of the combined data.

---

## Author Comment (AC1) · 19 Sep 2016

**Author General Comments**

We thank the referees for their comments and the time they spent for analyzing our paper and providing a detailed analysis. This helped to find a few deficiencies in our explanations and to improve our paper.

In the following part the referee comments are printed in italic letters, comments and answers are clearly marked with headers.

**Referee #1**

**Comment:** *For measurements of some quantity the instruments of different types are often used. Particularly in space-born projects as well as during ground-based geophysical surveys the magnetic field fluctuations are simultaneously measured by fluxgate and search coil magnetometers. The optimal combining data of both instruments is important task for obtaining high quality results with the best available signalnoise ratio (SNR). From this point of view the approach, methods, models and techniques described in the paper are important not only for a case study devoted to the Magnetosperic Multiscale mission (MMS), but for a wide range of similar applications. Surely the paper tries to answer questions within the scope of GI journal.*

*In general the approach used by authors to merge data from the instruments with different frequency responses is straightforward and clear. It contains following steps. Basing on the noise characteristics specify the merging frequency, where noise levels of both magnetometers are equal. Then estimate the end-to-end frequency responses of individual instruments and develop its appropriate models. Apply these models for the frequency compensation of the magnetometers data, convert data to the same sampling frequency and synchronize samplings. Make alignment of the axes and gain corrections and, finally, apply appropriate crossover filters and merge the data taken in the appropriate for each magnetometer frequency bands.*

*The presented in the paper example of this approach application confirms that merging in-flight data were synchronized within required limits (<100 $\mu s$), however the estimation of the gain mismatch was quite large due to the low signal-noise ratio in the analyzed frequency band 10-50 Hz. Probably, a higher signal-noise ratio and better validation of the proposed merging procedure could be obtained, if the records of the noise signals used for the frequency response calibration would be compared instead or in addition to the in-flight data processing.*

**Answer:** I agree that a higher gain accuracy is reached by using ground test data. However, such a test is generating a circular conclusion, as it only states that the model generation process based on our measurement process was correct, but does not verify the onboard merging.

We therefore chose to only include the inflight result, as this result shows the solution quality in an end to end configuration with an independent field source, but low SNR.

Furthermore during ground tests no common measurements of DFG and SCM were conducted, as DFG causes high frequency near field distortion for SCM in close proximity (like the test setup). A merged product of these tests would therefore

consist of 2 separate measurements with different noise in the shielding can. The error signals of models in these separate measurements were also analyzed, but did not deliver noteworthy gain/phase errors.

**Comment:** *The estimation of the noise level of the merged fluxgate and search coil data is highly recommended, as it would also confirm achieving of the main goal of this work – improving SNR of the combined data.*

**Answer:** Added a PSD plot of a quiet field region that shows the improved sensitivity of the merged product, but the noise floor is still covered by natural field. MMS team members are currently working on an improved noise floor estimation that will presumably be published within the next half year.

**Comment:** *In my opinion, the step and impulse responses of the merged data have to be checked too, in order to estimate the possible differences between the combined data and the original magnetic field signatures during the fast changes of the latter.*

**Answer:** The step and impulse response of merged data is not available for inflight data, as natural steps and impulses are not available. We can therefore only compare the compensated data of the instruments and their uncompensated version.

This analysis is done best in frequency domain, as it is hard to evaluate the differences in time domain. The higher frequency components will typically have very low amplitude and are not separatable from noise.
In fact we did time domain analysis during the first phase of implementation to get rid of bugs that caused large errors (millisecond range), but changed to frequency domain representation for more detailed analysis.

The difficulty of detecting small signal phase shifts is also visible in figure 11. It is impossible to tell if the small differences we see are due to noise or due to wrong higher harmonics with wrong phase or amplitude. A frequency domain analysis is giving more information and was therefore done both for single events as well as statistically (see figure 13)

**Comment:** *The important task is a selection of the proper parameters of the crossover filter for merging data. The authors states that a high order crossover filter is necessary in order "...to track the best noise floor..." of the instruments, however, this statement was not backgrounded in the paper and no appropriate references were given. First of all, it is unclear how the crossover filter could track the instruments' noise floor.*

**Answer:** changed the text to clarify. "Tracking" meant weighing different frequencies according to the onboard noise and distortion floor as well as sensitivity.

**Comment:** *Secondly, in my opinion, even the second order crossover filter is sufficient for combining data.*

**Answer:** A second order IIR crossover filter is in most cases an FIR filter of infinite order. The impulse response of an IIR (which is its FIR representation) can be found by long division of the numerator and denominator polynomials. This division will in

most cases not result in a finite series. The orders of FIR and IIR filters are therefore not directly comparable.

The choice of the FIR filter over the IIR was driven by the wish for constant group delay (allows comparison of the data before and after the merging filter) and an inherently known filter settling time without the need to analyze numerical properties. The cost of this choice was computational efficiency and the existence of passband ripple. This ripple is compensated by the inverse ripple of the respective opposite filter, but adds e.g. a little more SCM signal of a frequency band with lower quality. Steepness could be another cost, but this one is rather a question of the chosen crossover characteristics.

Nevertheless I agree that a 2$^{nd}$ order IIR filter is a valid solution, it just depends on the conditions. For a long data set and without the wish for comparison this would definitely be a solution with lower computational complexity.

**Comment:** *The overall presentation is quite well structured and clear, however, in some cases using mathematical expressions and formulae instead of text descriptions, as well as giving more details about the developed models and data processing techniques would be more useful for reproduction of the proposed approach by other researchers.*

Included the type of fit and IIR model used for the SCM. All the other techniques and models are considered standard signal processing used in literature and so only references were included.

**Specific comments**
**Comment:** *p. 6, section 2.3 Model Development.*
*For the sake of traceability of results the developed IIR model of SCM would be presented by expression in analytic form, and the typical shape of the impulse response of the FIR models for fluxgate magnetometers would be given.*
**Answer:** Added 2$^{nd}$ order formula and fit formula for SCM. Added a plot of typical AFG and DFG compensation filter impulse response.

**Comment:** *p. 4, lines 27-29 and p. 5, lines 3-7. Was the current in the coil measured by an independent instrument? If not, it is not clear how accurately the frequency response of the DSP voltage channels was estimated.*
**Answer:** The signal representative to the current was measured in the DSP voltage channel with sine input stimuli (page 4, second test). These measurements showed flat frequency and constant known delay in the frequency band of interest (0-1024Hz) for the pink noise test. As the voltage channels can operate up to 65 kHz, this was expected.

**Comment:** *Is it assumed that the current amplitude at all applied frequencies was constant?*
**Answer:** No, both because of the current source and because of the nature of the pink noise signal (low pass characteristics). Both problems were handled by using the current measurement as reference signal for the magnetic field rather than the desired current given by the control signal. Of course this does not exclude the possibility for an error in the current measurement and a frequency dependency of the current vs. magnetic field characteristics (e.g. caused by eddy currents in the can).
The comparison between reference measurements of the SCM team (le Contel et al, 2104) and our measurements show that no relevant differences are present, which leads to the conclusion that the assumption "current = field" was exact enough.

**Comment:** *p. 5 lines 14-22. What were parameters of FFT-based frequency response calibration - duration of the stimulus signals, FFT length, overlapping, type of the correcting window, if any?*
**Answer:** Changed figure caption to contain requested data.

**Comment:** *What are principal limitations, which did not allow improving phase delay estimations at the frequencies < 20 Hz?*
**Answer:** The limitation is the general noise level within the measurement. This noise level causes phase noise, which is then converted to delay noise. As phase delay is calculated by division with frequency (and 360°/2pi), a constant phase noise results in higher delay noise for lower frequencies.
Example: superposition of a 0 dB sine signal with a -80 dB cosine would cause a phase shift of $5 \times 10^{-3}$ deg. For a frequency of 0.1Hz this corresponds to a phase delay of 9 ms.

**Comment:** *p. 6, lines 12-13. It is unclear why so high DC gain (220 dB) of the shelving filter is necessary for frequency compensation of the SCM transfer function in the band from 0.1 Hz to 500 Hz, while the ratio of the gains of the original transfer function at these frequencies is approximately equal to -86 dB (Figure 4 in the manuscript and Figures 9 and 14 in Le Contel et al., 2014).*
**Answer:** The SCM transfer function has a gain drop of 40dB/decade for lower frequencies which ends at 0 (-∞ dB) for 0 Hz (DC). Since both Le Contel and we chose a logarithmic scaling, this part is not visible in the plots.
The inverse filter would have infinite gain at DC. The 220 dB are a tradeoff between compensating gain and phase behavior down to very low frequencies and having a realizable filter without numerical instabilities. With 220dB the compensation filter is roughly valid down to ~50 mHz in phase and 1 mHz in gain.

**Technical corrections**
**Comment:** *p. 2, line 4, in my opinion, it is better to use "...5 pT/√Hz..." or "...5 pT/Hz$_{1/2}$..." or "...5 pT Hz$_{-1/2}$..." instead of "...5 pT √Hz$_{-1}$..."*
**Answer:** I changed it to pT Hz$^{-1/2}$ and hope this meets publisher requirements. The first suggestion would have been my favorite style, but seems to be against requirements.

**Comment:** *p. 4, line 31, probably word "with" could be omitted.*
**Answer:** Removed in draft for next version.

**Comment:** *p. 5, lines 7, 8. The sentence "The later noise tests were therefore conducted using the current measurements of the DSP channels, as this resulted in reduced effort in calculations." What does the phrase "the current measurements of the DSP channels" mean? Does it mean that during the noise tests the voltage proportional to the stimulus current in the coil was measured using one of the DSP voltage channels? In this case, probably, it has to be formulated as follows "the current measurements **via** the DSP channels".*
**Answer:** Assumption is correct. Changed to "via"

**Comment:** *p. 6, line 26, probably, the second word "instead" could be omitted.*
**Answer:** Removed in draft for next version.

**Comment:** *p. 9, lines 17, 18, probably, the second word "interpretation" could be omitted.*

**Answer:** Removed in draft for next version.

**Comment:** *The combining data which is simultaneously measured by fluxgate and search coil magnetometers is important task for obtaining high quality results. So the tasks which authors solved are important for different applications. After correction the method of merging of search coil and fluxgate data became clear. Authors confirm that merging in-flight data can be synchronized within 100 mks. The unclear question is the noise level of the combined data.*
**Answer:** Identical to 2nd comment of referee #1. Added a PSD plot and text.